# First Trimester Prediction of Preterm Delivery in the Absence of Other Pregnancy-Related Complications Using Cardiovascular-Disease Associated MicroRNA Biomarkers

**DOI:** 10.3390/ijms23073951

**Published:** 2022-04-01

**Authors:** Ilona Hromadnikova, Katerina Kotlabova, Ladislav Krofta

**Affiliations:** 1Department of Molecular Biology and Cell Pathology, Third Faculty of Medicine, Charles University, 100 00 Prague, Czech Republic; katerina.kotlabova@lf3.cuni.cz; 2Institute for the Care of the Mother and Child, Third Faculty of Medicine, Charles University, 147 00 Prague, Czech Republic; ladislav.krofta@upmd.eu

**Keywords:** cardiovascular microRNAs, early gestation, expression, prediction, preterm delivery, preterm prelabor rupture of membranes, screening, spontaneous preterm birth, whole peripheral venous blood

## Abstract

The aim of the study was to determine if aberrant expression profile of cardiovascular disease associated microRNAs would be able to predict within 10 to 13 weeks of gestation preterm delivery such as spontaneous preterm birth (PTB) or preterm prelabor rupture of membranes (PPROM) in the absence of other pregnancy-related complications (gestational hypertension, preeclampsia, fetal growth restriction, or small for gestational age). In addition, we assessed if aberrant expression profile of cardiovascular disease associated microRNAs would be able to predict preterm delivery before and after 34 weeks of gestation. The retrospective study was performed within the period November 2012 to March 2020. Whole peripheral blood samples were collected from 6440 Caucasian individuals involving 41 PTB and 65 PPROM singleton pregnancies. A control group, 80 singleton term pregnancies, was selected on the base of equal sample storage time. Gene expression of 29 selected cardiovascular disease associated microRNAs was studied using real-time RT-PCR. Downregulation of miR-16-5p, miR-20b-5p, miR-21-5p, miR-24-3p, miR-26a-5p, miR-92a-3p, miR-126-3p, miR-133a-3p, miR-145-5p, miR-146a-5p, miR-155-5p, miR-210-3p, miR-221-3p and miR-342-3p was observed in pregnancies with preterm delivery before 37 (≤36 + 6/7) weeks of gestation. Majority of downregulated microRNAs (miR-16-5p, miR-24-3p, miR-26a-5p, miR-92a-3p, miR-133a-3p, miR-145-5p, miR-146a-5p, miR-155-5p, miR-210-3p, and miR-342-3p) was associated with preterm delivery occurring before 37 (≤36 + 6/7) weeks of gestation. The only miR-210-3p was downregulated in pregnancies with preterm delivery before 34 (≤33 + 6/7) weeks of gestation. The type of preterm delivery also had impact on microRNA gene expression profile. Downregulation of miR-24-3p, miR-92a-3p, miR-155-5p, and miR-210-3p was a common feature of PTB and PPROM pregnancies. Downregulation of miR-16-5p, miR-20b-5p, miR-26a-5p, miR-126-3p, miR-133a-3p, miR-146a-5p, miR-221-3p, and miR-342-3p appeared just in PTB pregnancies. No microRNA was uniquely dysregulated in PPROM pregnancies. The combination of 12 microRNAs (miR-16-5p, miR-20b-5p, miR-21-5p, miR-24-3p, miR-26a-5p, miR-92a-3p, miR-133a-3p, miR-145-5p, miR-146a-5p, miR-155-5p, miR-210-3p, and miR-342-3p, AUC 0.818, *p* < 0.001, 74.53% sensitivity, 75.00% specificity, cut off > 0.634) equally as the combination of 6 microRNAs (miR-16-5p, miR-21-5p, miR-24-3p, miR-133a-3p, miR-155-5p, and miR-210-3p, AUC 0.812, *p* < 0.001, 70.75% sensitivity, 78.75% specificity, cut off > 0.652) can predict preterm delivery before 37 weeks of gestation in early stages of gestation in 52.83% pregnancies at 10.0% FPR. Cardiovascular disease associated microRNAs represent promising biomarkers with very good diagnostical potential to be implemented into the current routine first trimester screening programme to predict preterm delivery.

## 1. Introduction

Spontaneous preterm birth (PTB) is defined as the delivery before completed 37 week of gestation caused by regular uterine contractions along with cervical changes. Preterm prelabor rupture of membranes (PPROM), rupture of membranes (ROM) prior to 37 weeks of gestation, is characterized as amniotic fluid leakage preceding the onset of labour by at least 2 h [1,2,3].

Preterm delivery in anamnesis of women was demonstrated to be associated with an increased cardiovascular risk in young and middle-aged women [4,5,6,7,8,9,10,11,12,13,14,15,16,17,18,19,20,21,22,23,24,25,26,27].

Recently, we have demonstrated that mothers after pregnancies complicated with PPROM or PTB in the absence of other pregnancy-related complications such as gestational hypertension (GH), preeclampsia (PE), fetal growth restriction (FGR), or small for gestational age (SGA) had aberrant expression profile of overwhelming majority from tested cardiovascular disease associated microRNAs [4]. MiR-1-3p, miR-16-5p, miR-17-5p, miR-20a-5p, miR-20b-5p, miR-21-5p, miR-23a-3p, miR-24-3p, miR-26a-5p, miR-29a-3p, miR-100-5p, miR-103a-3p, miR-125b-5p, miR-126-3p, miR-130b-3p, miR-133a-3p, miR-143-3p, miR-145-5p, miR-146-5p, miR-181a-5p, miR-195-5p, miR-199a-5p, miR-221-3p, miR-499a-5p, and miR-574-3p showed significantly increased postpartum expression in whole peripheral blood samples of mothers with a history of PPROM or PTB pregnancies [4].

Currently, we were interested if the dysregulation of gene expression of cardiovascular disease associated microRNAs would be already present in the whole peripheral venous blood during the early stages of gestation and would be able to predict preterm delivery (PTB or PPROM) in the absence of other pregnancy-related complications.

We were also interested if the first trimester profiling of cardiovascular disease associated microRNAs in the whole peripheral venous blood would have been able to differentiate between preterm and term pregnancies with respect to the gestational age of the preterm delivery (before and after 34 weeks of gestation).

To date, little data on the first trimester microRNA profiling in the whole peripheral venous blood is available in pregnancies with preterm delivery.

Winger et al. achieved reliable first trimester prediction of PTB via quantification of microRNAs in peripheral blood mononuclear cells using real-time RT-PCR [28,29]. Initially, Winger et al. [28] tested a panel of 30 microRNAs (miR-340-5p, miR-424-5p, miR-33a-5p, miR-7-5p, miR-1229, miR-1267, miR-671-3p, miR-1, miR-133b, miR-144-3p, miR-582-5p, miR-30e-3p, miR-199a-5p, miR-199b-5p, miR-210, miR-221-5p, miR-575, miR-301a-3p, miR-148a-3p, miR-193a-3p, miR-219-5p, miR-132, miR-513a-5p, miR-1244, miR-16, miR-146a, miR-155, miR-181a, miR-196a, and miR-223) and later introduced a minimal panel of 8 microRNAs (miR-340-5p, miR-1267, miR-671-3p, miR-210, miR-301a-3p, miR-148a-3p, miR-181a, and miR-223) to determine the risk of PTB during the first trimester of gestation in a small-scale study involving 14 PTB and 25 term pregnancies (AUC 0.95, *p* < 0.001).

In addition, Winger et al. [29] tested an extended panel of 45 microRNAs (miR-340-5p, miR-424-5p, miR-33a-5p, miR-7-5p, miR-1229, miR-1267, miR-671-5p, miR-1-3p, miR-133b, miR-144-3p, miR-582-5p, miR-30e-3p, miR-30e-5p, miR-199a-5p, miR-199b-5p, miR-210-3p, miR-210-5p, miR-221-3p, miR-221-5p, miR-575, miR-301a-3p, miR-148a-3p, miR-193a-3p, miR-219-5p, miR-132-3p, miR-513-5p, miR-1244, miR-16-5p, miR-146a-5p, miR-155-5p, miR-181a-3p, miR-181a-5p, miR-196a-5p, miR-223-5p, miR-223-3p, miR-6752-3p, miR-4485-5p, miR-1237-3p, miR-6889-3p, miR-551b-3p, miR-6819-3p, miR-1238-3p, miR-6737-3p, miR-24-1-5p, and miR-6757-3p) and subsequently proposed a minimal panel of 12 microRNAs (miR-33a-5p, miR-1267, miR-1-3p, miR-133b, miR-144-3p, miR-199b-5p, miR-221-3p, miR-148a-3p, miR-1244, miR-181a-3p, miR-223-5p, and miR-6752-3p) to assess the risk of PTB during the early stages of gestation in a small-scale study involving 9 PTB and 69 term pregnancies (AUC 0.80, *p* < 0.001).

An algorithm for the prediction of spontaneous preterm delivery in the first trimester of gestation (between 11 and 13 weeks) based on the combination of maternal characteristics, obstetrics history, biochemical markers such as free beta human chorionic gonadotrophin (beta-hCG), pregnancy-associated plasma protein-A (PAPP-A), placental growth factor (PIGF), placental protein 13 (PP13), a disintegrin and metalloprotease 12 (ADAM12), inhibin-A, activin-A, and Doppler ultrasound parameter (mean uterine artery pulsatility index) was not superior over a model based on maternal factors and obstetrics history only [30]. The screening based on maternal factors and obstetrics history was able to identify at 10.0% FPR spontaneous early preterm delivery in about 20% nulliparous women and 38% multiparous women [30]. This model was integrated into more complex Astraia Obstetrics programme, which has been currently used by most Fetal Medicine Centres to calculate the risks for trisomy 21, 18 and 13, preeclampsia, and fetal growth restriction during the first trimester of gestation [31,32,33,34,35].

## 2. Results

### 2.1. Clinical Characteristics of Patients and Controls

The case-control clinical characteristics are summarized in Table 1.

At first, microRNA gene expression in peripheral blood leukocytes was compared in early stages of gestation (within 10 to 13 weeks) between pregnancies with preterm (before 37 weeks of gestation) and term (after 37 weeks of gestation) delivery. Only singleton pregnancies with prelabor rupture of membranes or spontaneous preterm birth with an absence of other pregnancy-related complications such as gestational hypertension, preeclampsia, fetal growth restriction, and small for gestational age, were included in the study. Afterwards, early microRNA gene expression was compared between pregnancies with preterm and term delivery with respect to the occurrence of preterm delivery before or after 34 weeks of gestation. Finally, the difference in microRNA gene expression profiles in early stages of gestation was compared between preterm and term pregnancies with respect to the type of preterm delivery (preterm prelabor rupture of membranes or spontaneous preterm birth). Only the data that reached statistical significance after the application of Benjamini-Hochberg correction for multiple comparisons are discussed below (Appendix A). To interpret the experimental data new cut-off point *p*-values were set-up. Significant results after the Benjamini-Hochberg correction are marked by asterisks for appropriate significance levels (* for α = 0.05, ** for α = 0.01, and *** for α = 0.001). The data that were statistical non-significant after the application of Benjamini-Hochberg correction are also displayed (Appendix A), but not discussed further.

### 2.2. Dysregulated Expression Profile of Cardiovascular Disease Associated MicroRNAs in Early Stages of Gestation Is Able to Predict Preterm Delivery

The gene expression of miR-16-5p (*p* = 0.004 **), miR-20b-5p (*p* = 0.009 *), miR-21-5p (*p* = 0.013 *), miR-24-3p (*p* < 0.001 ***), miR-26a-5p (*p* = 0.003 **), miR-92a-3p (*p* < 0.001 ***), miR-126-3p (*p* = 0.009 *), miR-133a-3p (*p* = 0.001 **), miR-145-5p (*p* = 0.004 **), miR-146a-5p (*p* = 0.003 **), miR-155-5p (*p* < 0.001 ***), miR-210-3p (*p* < 0.001 ***), miR-221-3p (*p* = 0.018 *), and miR-342-3p (*p* < 0.001 ***) was downregulated in early stages of gestation in pregnancies with preterm delivery (before 37 weeks of gestation) when compared with normal term pregnancies (after 37 weeks of gestation) (Appendix A).

The ROC curve analyses showed the following sensitivities at 10.0% FPR for miR-16-5p (29.25%), miR-20b-5p (21.70%), miR-21-5p (33.02%), miR-24-3p (28.30%), miR-26a-5p (23.58%), miR-92a-3p (22.64%), miR-126-3p (11.32%), miR-133a-3p (29.25%), miR-145-5p (18.87%), miR-146a-5p (22.64%), miR-155-5p (22.64%), miR-210-3p (34.91%), miR-221-3p (13.21%), and miR-342-3p (25.47%) in preterm pregnancies (Appendix A).

MiR-126-3p (11.32%) and miR-221-3p (13.21%) showed poor sensitivities in preterm pregnancies at 10.0% FPR, therefore these particular microRNAs were not further used to identify the best microRNA combination.

The combination of 12 microRNAs (miR-16-5p, miR-20b-5p, miR-21-5p, miR-24-3p, miR-26a-5p, miR-92a-3p, miR-133a-3p, miR-145-5p, miR-146a-5p, miR-155-5p, miR-210-3p, and miR-342-3p) was superior over using other combinations of microRNAs or individual microRNA biomarkers. It was able to detect at 10.0% FPR in early stages of gestation aberrant microRNA expression profile in 52.83% pregnancies with preterm delivery (AUC 0.818, *p* < 0.001, 74.53% sensitivity, 75.00% specificity, cut off > 0.634) (Figure 1). Variance inflating factor (VIF) was used to test the presence of multicollinearity in a regression model. All 12 tested microRNAs had VIF below 10 (miR-16-5p, 5.410; miR-20b-5p, 2.186; miR-21-5p, 5.337; miR-24-3p, 5.077; miR-26a-5p, 5.396; miR-92a-3p, 5.526; miR-133a-3p, 1.090; miR-145-5p, 4.077; miR-146a-5p, 5.129; miR-155-5p, 3.039; miR-210-3p, 2.924; and miR-342-3p, 2.093) and therefore were involved in the combination.

The combination of 6 microRNAs (miR-16-5p, miR-21-5p, miR-24-3p, miR-133a-3p, miR-155-5p, and miR-210-3p) represented the combination of minimal number of microRNA biomarkers to achieve similar accuracy to predict the occurrence of preterm delivery. It was able to detect at 10.0% FPR in early stages of gestation aberrant microRNA expression profile in 52.83% pregnancies with preterm delivery (AUC 0.812, *p* < 0.001, 70.75% sensitivity, 78.75% specificity, cut off > 0.652) (Figure 2). Again, VIF was used to test the presence of multicollinearity in a regression model. All 6 tested microRNAs had VIF below 10 (miR-16-5p, 4.900; miR-21-5p, 4.153; miR-24-3p, 5.407; miR-133a-3p, 1.078; miR-155-5p, 2.327; and miR-210-3p, 2.611) and therefore were involved in the combination.

### 2.3. Prediction Model of Preterm Delivery Based on the Combination of MicroRNA Biomarkers, Maternal Age at Sampling and Serum PAPP-A Levels (IU/L)

The combination of 12 microRNAs (miR-16-5p, miR-20b-5p, miR-21-5p, miR-24-3p, miR-26a-5p, miR-92a-3p, miR-133a-3p, miR-145-5p, miR-146a-5p, miR-155-5p, miR-210-3p, and miR-342-3p), maternal age at sampling and serum PAPP-A levels (IU/L) provided similar predictive results as the combination of 12 microRNA biomarkers only. It was able to predict at 10.0% FPR in early stages of gestation the occurrence of preterm delivery in 50.94% pregnancies (AUC 0.824, *p* < 0.001, 76.42% sensitivity, 75.95% specificity, cut off >0.637) (Figure 3).

The combination of 6 microRNAs (miR-16-5p, miR-21-5p, miR-24-3p, miR-133a-3p, miR-155-5p, and miR-210-3p), maternal age at sampling and serum PAPP-A levels (IU/L) provided similar predictive results as the combination of 6 microRNA biomarkers only. It was able to predict at 10.0% FPR in early stages of gestation the occurrence of preterm delivery in 49.06% pregnancies (AUC 0.812, *p* < 0.001, 77.36% sensitivity, 70.89% specificity, cut off > 0.608) (Figure 4).

### 2.4. Dysregulated Expression Profile of Cardiovascular Disease Associated MicroRNAs in Early Stages of Gestation with Respect to the Gestational Age of the Delivery

The gene expression of miR-210-3p (*p* < 0.009 *, *p* < 0.001 ***) was downregulated during the first trimester of gestation in pregnancies with preterm delivery regardless the gestational age of the delivery. Decreased levels of miR-16-5p (*p* = 0.015 *), miR-24-3p (*p* < 0.001 ***), miR-26a-5p (*p* = 0.007 *), miR-92a-3p (*p* = 0.002 **), miR-133a-3p (*p* = 0.004 *), miR-145-5p (*p* = 0.004 *), miR-146a-5p (*p* = 0.006 *), miR-155-5p (*p* < 0.001 ***), and miR-342-3p (*p* = 0.004 *) were detected in pregnant women with preterm delivery after 34 weeks of gestation only (Appendix A).

The ROC curve analyses showed very good sensitivities at 10.0% FPR for miR-16-5p (32.47%), miR-24-3p (32.47%), miR-26a-5p (24.68%), miR-92a-3p (22.08%), miR-133a-3p (29.87%), miR-145-5p (19.48%), miR-146a-5p (25.97%), miR-155-5p (25.97%), miR-210-3p (36.36%), and miR-342-3p (27.27%) in pregnancies with preterm delivery after 34 weeks of gestation (Appendix A).

Downregulation of miR-210-3p was also present in early stages of gestation in 31.03% of pregnancies with preterm delivery before 34 weeks of gestation at 10.0% FPR, (Appendix A).

No relation between microRNA gene expression and gestational age at birth in patients with preterm delivery (PPROM and PTB) was observed (Appendix A).

### 2.5. Dysregulated Expression Profile of Cardiovascular Disease Associated MicroRNAs in Early Stages of Gestation with Respect to the Type of Preterm Delivery

The gene expression of miR-16-5p (*p* = 0.009 *), miR-20b-5p (*p* < 0.001 **), miR-26a-5p (*p* = 0.012 *), miR-126-3p (*p* = 0.006*), miR-133a-3p (*p* = 0.002 **), miR-146a-5p (*p* = 0.006 *), miR-221-3p (*p* < 0.010 *), and miR-342-3p (*p* = 0.001 **) was downregulated in PTB pregnancies only (Appendix A).

Decreased levels of miR-24-3p (*p* = 0.001 **, *p* = 0.008 *), miR-92a-3p (*p* = 0.002 **, *p* = 0.022 *), miR-155-5p (*p* = 0.004 **, *p* = 0.002 **), and miR-210-3p (*p* < 0.001 ***, *p* < 0.001 ***) were observed during early stages of gestation in preterm pregnancies irrespective of the type of delivery (PTB or PPROM) (Appendix A).

The ROC curve analyses showed very good sensitivities at 10.0% FPR for miR-16-5p (36.59%), miR-20b-5p (34.15%), miR-26a-5p (26.83%), miR-133a-3p (36.59%), miR-146a-5p (26.83%), miR-221-3p (19.51%), and miR-342-3p (31.71%) in PTB pregnancies.

A significant proportion of PTB and PPROM pregnancies had downregulated expression profile of miR-24-3p (34.15%, 24.62%), miR-92a-3p (34.15%, 15.38%), miR-155-5p (17.07%, 26.15%), and miR-210-3p (41.46%, 30.77%) at 10.0% FPR during the first trimester of gestation (Appendix A).

MiR-126-3p (12.20%) showed the poor sensitivity in PTB pregnancies at 10.0% FPR, therefore this particular microRNA was not further considered as early gestation biomarker distinguishing between normal and pathological course of gestation (Appendix A).

The combination of 9 microRNAs (miR-16-5p, miR-20b-5p, miR-24-3p, miR-26a-5p, miR-92a-3p, miR-133a-3p, miR-146a-5p, miR-210-3p, and miR-342-3p) was superior to predict spontaneous preterm birth in early stages of gestation. It was able to detect at 10.0% FPR in early stages of gestation aberrant microRNA expression profile in 43.90% PTB pregnancies (AUC 0.803, *p* < 0.001, 73.17% sensitivity, 76.25% specificity, cut off > 0.428) (Figure 5). VIF was used to test the presence of multicollinearity in a regression model. All 9 tested microRNAs had VIF below 10 (miR-16-5p, 5.634; miR-20b-5p, 2.093; miR-24-3p, 5.398; miR-26a-5p, 5.629; miR-92a-3p, 5.621; miR-133a-3p, 1.082; miR-146a-5p, 5.359; miR-210-3p, 2.643; and miR-342-3p, 5.110) and therefore were involved in the combination.

Similarly, the combination of 4 microRNAs (miR-24-3p, miR-92a-3p, miR-155-5p, and miR-210-3p) showed the best accuracy to predict the occurrence of preterm prelabor rupture of membranes. It was able to detect at 10.0% FPR in early stages of gestation aberrant microRNA expression profile in 30.77% PPROM pregnancies (AUC 0.720, *p* < 0.001, 78.46% sensitivity, 58.75% specificity, cut off > 0.473) (Figure 6). Again, VIF was used to test the presence of multicollinearity in a regression model. All 4 tested microRNAs had VIF below 10 (miR-24-3p, 5.901; miR-92a-3p, 3.868; miR-155-5p, 2.534; and miR-210-3p, 2.582) and therefore were involved in the combination.

### 2.6. Information on MicroRNA-Gene-Biological Pathways Interactions

Information on microRNA-gene-biological pathways interactions was provided on microRNAs dysregulated in the whole peripheral blood of pregnancies at risk of preterm delivery. Predicted targets of microRNAs involved in key human biological pathways, with a role in the pathogenesis of preterm delivery, were reported. These biological pathways involve apoptosis, inflammatory response, senescence, and autophagy.

A large group of genes (predicted targets) of cardiovascular disease associated microRNAs observed to be downregulated in the whole peripheral blood during the first trimester of gestation in pregnancies at risk of preterm delivery is also involved in key biological processes related to the pathogenesis of preterm delivery, such as the apoptosis pathway, the inflammatory response pathway, the senescence and autophagy pathways (Table 2 and Table 3).

## 3. Discussion

Gene expression of cardiovascular disease associated microRNAs was tested in peripheral blood leukocytes during the early stages of gestation (within 10 to 13 weeks) with the aim to distinguish between preterm pregnancies (PPROM or PTB) and term pregnancies in the absence of other pregnancy-related complications such as GH, PE, FGR or SGA. In addition, early microRNA gene expression profile was compared between pregnancies with preterm delivery before and after 34 weeks of gestation and pregnancies with term delivery after 37 weeks of gestation.

Overall, downregulation of 14 out of 29 tested microRNAs (miR-16-5p, miR-20b-5p, miR-21-5p, miR-24-3p, miR-26a-5p, miR-92a-3p, miR-126-3p, miR-133a-3p, miR-145-5p, miR-146a-5p, miR-155-5p, miR-210-3p, miR-221-3p and miR-342-3p) was observed in early stages of gestation in pregnancies with preterm delivery before 37 weeks of gestation. Finally, minimal combination of 6 selected microRNAs (miR-16-5p, miR-21-5p, miR-24-3p, miR-133a-3p, miR-155-5p, and miR-210-3p) or maximal combination of 12 selected microRNAs (miR-16-5p, miR-20b-5p, miR-21-5p, miR-24-3p, miR-26a-5p, miR-92a-3p, miR-133a-3p, miR-145-5p, miR-146a-5p, miR-155-5p, miR-210-3p, and miR-342-3p) was sufficient enough to predict the occurrence of preterm delivery in 52.83% cases during the first trimester of gestation.

Interestingly, majority of first trimester downregulated microRNAs (miR-16-5p, miR-24-3p, miR-26a-5p, miR-92a-3p, miR-133a-3p, miR-145-5p, miR-146a-5p, miR-155-5p, miR-210-3p, and miR-342-3p) was associated with the occurrence of preterm delivery after 34 weeks of gestation. The only miR-210-3p was downregulated in early stages of gestation in relation to the onset of preterm delivery before 34 weeks of gestation.

The microRNA gene expression profile also differed significantly during the early stages of gestation in peripheral blood leukocytes with respect to the type of preterm delivery. Downregulation of miR-24-3p, miR-92a-3p, miR-155-5p, and miR-210-3p was a common feature of PTB and PPROM pregnancies. Downregulation of miR-16-5p, miR-20b-5p, miR-26a-5p, miR-126-3p, miR-133a-3p, miR-146a-5p, miR-221-3p, and miR-342-3p occurred in early stages of gestation just in PTB pregnancies. No microRNA was uniquely dysregulated in PPROM pregnancies.

Our data are consistent with the data of Winger et al. [28,29], who also observed dysregulation of miR-16-5p, miR-146a-5p, miR-155-5p, miR-210-3p, and miR-221-3p in peripheral blood mononuclear cells in PTB pregnancies during the first trimester of gestation. Nevertheless, concerning miR-1-3p, miR-181a-5p, and miR-199a-5p our data are inconsistent with the data of Winger at al. [28,29]. While we observed no difference in microRNA gene expression between preterm and term pregnancies in peripheral blood leukocytes, they observed aberrant microRNA expression levels in peripheral blood mononuclear cells in PTB pregnancies during the early stages of gestation and implemented miR-1-3p or miR-181a-5p into the panels of microRNAs to predict later occurrence of PTB.

Dysregulation of most microRNAs (miR-16-5p, miR-20b-5p, miR-24-3p, miR-26a-5p, miR-126-3p, miR-133a-3p, miR-146a-5p, miR-221-3p, and miR-342-3p) was also observed postpartally in women with a history of PPROM or PTB pregnancies in the absence of other pregnancy-related complications [4], however postpartum microRNA gene expression was significantly increased in peripheral blood leukocytes, whereas during the early stages of gestation microRNA gene expression was downregulated.

Interestingly, in addition, dysregulation of some microRNAs (miR-92a-3p, miR-155-5p, and miR-210-3p) appeared exclusively just during the first trimester of gestation, however disappeared in postpartum period in women with a history of preterm delivery [4]. This finding implies that microRNA gene expression profile may differ between prenatal and postpartum periods, but it is evident that aberrant microRNA gene expression profile is mostly present all the time and indicates a potential association with a complicated course of gestation, this time in a form of preterm delivery.

Recently, we have observed during the first trimester of gestation upregulation of miR-146a-5p and miR-155-5p in pregnancies with chronic hypertension and normotensive pregnancies with subsequent onset of PE, FGR or SGA [37,38]. Current study revealed that the expression of miR-146a-5p and miR-155-5p is downregulated during the early stages of gestation in pregnancies with preterm delivery.

Parallel, not long ago we observed upregulation of miR-20b-5p and miR-126-3p during the early stages of gestation in SGA pregnancies [38]. At present, dysregulation of miR-20b-5p and miR-126-3p was detected in preterm delivery, however the levels of microRNAs were significantly decreased.

Alongside, miR-145-5p upregulation appeared in early stages of gestation in normotensive pregnancies subsequently developing PE or FGR [37,38]. Currently, in pregnancies affected with preterm delivery, downregulation of miR-145-5p was observed.

At the same time, upregulation of miR-16-5p and miR-342-3p was present in early stages of gestation in normotensive pregnancies with subsequent onset of FGR and besides upregulation of miR-16-5p was also found in SGA pregnancies [38]. Concurrently, downregulation of miR-16-5p and miR-342-3p was observed to be associated with preterm delivery.

On the other hand, downregulation of miR-21-5p, miR-24-3p, miR-26a-5p, miR-92a-3p, miR-133a-3p, miR-210-3p, and miR-221-3p in early gestational stages represents a unique feature of preterm delivery.

The pathogenesis of preterm delivery is multifactorial, but some principal pathogenic mechanisms such as cervical insufficiency, uterine malformations, acute inflammation of the membranes and chorion of the placenta followed by an exaggeration of inflammatory processes have already been discovered [3,39,40,41,42,43,44]. Moreover, premature aging of fetal membranes encompasses telomere shortening, senescence, apoptosis, and proteolysis, had also been demonstrated to play a key role in pathogenesis of preterm delivery [45,46,47]. From the analyses on microRNA-gene-biological pathways interactions it is evident that microRNAs with altered expression profiles in mothers at risk of preterm delivery during the first trimester of gestation interact with specific genes involved in key biological processes related to the pathogenesis of preterm delivery, such as the apoptosis pathway, the inflammatory response pathway, the senescence and autophagy pathways. In addition, the expression of series of genes involved in key biological processes related to the pathogenesis of preterm delivery was concurrently observed to be dysregulated during the onset of spontaneous preterm birth in maternal peripheral blood within 27-36 weeks of gestation [36]. Meta-analysis of maternal and fetal transcriptomic data revealed in total 210 differentially expressed genes (65 genes upregulated and 145 genes downregulated) in maternal peripheral blood during the third trimester of gestation in PTB pregnancies, in which case half of these genes were immune related. Dysregulated genes were highly involved in the innate and adaptive immune responses [36]. For example, within upregulated genes IL1R1, producing IL-1 receptor type I (IL-1R1) [36], whose expression may be post-transcriptionally regulated by various microRNAs, inclusive of miR-20b-5p, was identified. Similarly, within upregulated genes IL1RAP, producing IL-1 receptor accessory protein [36], whose expression may be post-transcriptionally regulated by various microRNAs, inclusive of miR-221-3p, was detected. In addition, within upregulated genes IRAK3, producing IL-1 receptor associated Kinase 3 [36], whose expression may be post-transcriptionally regulated by various microRNAs, inclusive of miR-133a-3p, was found. On the other hand, within downregulated genes PMAIP1, producing phorbol-12-myristate-13-acetate-induced protein 1 [36], a pro-apoptotic member of the Bcl-2 protein family, whose expression may be post-transcriptionally regulated by various microRNAs, inclusive of miR-26a-5p, miR-146a-5p and miR-342-3p, was identified. Parallel, within downregulated genes CD28 (Cluster of Differentiation 28) [36], protein expressed on T cells providing costimulatory signals required for T cell activation and survival, whose expression may be post-transcriptionally regulated by various microRNAs, inclusive of miR-20b-5p, miR-24-3p, miR-92a-3p, miR-133a-3p, miR-155-5p, and miR-210-3p was detected. In addition, within downregulated genes TFDP2 (Transcription factor Dp-2) [36], whose expression may be post-transcriptionally regulated by various microRNAs, inclusive of miR-92a-3p and miR-210-3p, was found. This finding indicates that the foundations of the pathological course of gestation are laid already in the beginning of gestation.

Nevertheless, in relation to gestation, microRNA expression patterns have been changing from the first trimester to the third trimester and to the delivery, both in the placenta and maternal circulation. For example, a comparison between placentas isolated from the first and the third trimesters highlighted a total of 208 microRNA transcripts that were differentially expressed [48,49]. For that reason microRNA expression patterns in maternal circulation may also differ between the first trimester and the third trimester of gestation, the delivery, or postpartum period as an actual response reflecting the changes occurring either in the placenta or overall in the human body. As a consequence of these events, downregulation of certain microRNAs observed in pregnancies at risk of preterm delivery in the first trimester of gestation may be substituted by upregulation of the same microRNAs during the onset of preterm delivery, and vice versa. In addition, altered microRNA expression profile is afterwards in postpartum period normalized or in some women with a history of preterm delivery has been persisting long-term [4].

It seems that microRNA biomarkers might be implemented into the current first trimester screening programme to predict the occurrence of preterm delivery. Nevertheless, consecutive large-scale studies are needed to verify the potential of microRNA biomarkers to predict the onset of preterm delivery.

## 4. Materials and Methods

### 4.1. Patients Cohort

This retrospective study was held within the period 11/2012–3/2020 and involved singleton Caucasian pregnancies only. The whole peripheral blood samples were collected from 10 to 13 weeks of gestation in the course of the first trimester screening. In total, whole peripheral blood samples were collected from 6440 pregnancies. Finally, 4469 pregnancies had complete medical records, since they had been regularly followed-up and delivered in the Institute for the Care of Mother and Child, Prague, Czech Republic. Forty-one out of 4469 pregnancies were confirmed to have diagnosis of spontaneous preterm birth, and 65 pregnancies were diagnosed with preterm prelabor rupture of membranes. Only preterm pregnancies (delivery before 37 weeks of gestation) with the absence of other pregnancy-related complications (GH, PE, FGR, or SGA) were included in this case-control study. In detail, 29 pregnancies delivered before 34 weeks of gestation (≤33 + 6/7 weeks of gestation) and 77 pregnancies after 34 weeks of gestation (between 34 + 0/7 weeks to 36 + 6/7 weeks of gestation).

Term pregnancies with normal course of gestation that delivered healthy infants after 37 completed weeks of gestation with the weight over 2500 g were used as a control group (*n* = 80). Term normal pregnancies were selected on the basis of equal gestational age at the time of collection of whole peripheral blood samples, and equal biological sample storage times.

Informed consent with the study was gained from patients during the first trimester of gestation when the collection of peripheral blood samples for the first trimester screening was held. Informed consent was signed by all pregnant women involved in the study. The approval with the study was gained initially from the Ethics Committee of the Third Faculty of Medicine, Charles University (Implication of placental specific microRNAs in maternal circulation for diagnosis and prediction of pregnancy-related complications, date of approval: 7 April 2011). Ongoing approvals with the study were gained from the Ethics Committee of the Third Faculty of Medicine, Charles University (Long-term monitoring of complex cardiovascular profiles in mother, fetus, and offspring descending from pregnancy-related-complications, dates of approval: 27 March 2014) and the Ethics Committee of the Institute for the Care of the Mother and Child, Charles University (Long-term monitoring of complex cardiovascular profiles in mother, fetus, and offspring descending from pregnancy-related-complications, dates of approval: 28 May 2015, number of approval: 1/4/2015). This informed consent is very complex and involves the consent with the collection of peripheral blood samples at the beginning of pregnancy. In addition, in case of the onset of pregnancy-related complications (gestational hypertension, preeclampsia, FGR, and preterm delivery) it also involves the consent with the collection of peripheral blood samples at the time of the onset of pregnancy-related complications and the collection of a piece of placenta sample during the childbirth. All procedures were in compliance with the Helsinki Declaration of 1975, as revised in 2000.

### 4.2. Combined First Trimester Risk Analysis

The algorithm for the calculation of risks for trisomy 21, trisomy 18, trisomy 13, FGR, PE, and preterm delivery in the first trimester of gestation was produced by Astraia software GmbH, Germany (Astraia Obstetrics Programme) in close collaboration with the Fetal Medicine Foundation [35].

In our group of 106 preterm pregnancies 10 women were identified to be at risk of development of PE and/or FGR. Based on the ACOG 2018 guidelines [50] and NICE 2019 guidelines [51] low dose aspirin (ASA) was finally given to 7 out of 10 pregnant women. ASA (100 mg) was administered daily in the evening from 12th week of gestation (before 16th week of gestation at latest) to decrease the risk of development of PE and/or FGR. Finally, none of these pregnancies developed PE or FGR. In our control group of selected term pregnancies no pregnancy was identified to be at risk of development of PE and/or FGR and received no ASA.

In addition, in our group of 106 preterm pregnancies 25 women were identified to be at a risk of preterm delivery before 34 weeks of gestation at a cut-off value 1:100, which represents 23.58% cases. Four of these 25 pregnancies had simultaneously first trimester screening positive for PE and/or FGR. ASA was finally administered to 2 out of these 4 pregnant women. Simultaneously, 5 out of 80 control term pregnancies (6.25%) were identified false positively to be at risk of preterm delivery before 34 weeks of gestation.

### 4.3. Processing of Samples

Homogenized leukocyte lysates were prepared from 200 µL maternal whole peripheral venous blood samples immediately after collection using QIAamp RNA Blood Mini Kit (Qiagen, Hilden, Germany) according to manufacturer’s instructions. Firstly, lysis of erythrocytes was performed using EL buffer and then pelleted leukocytes were stored in a mixture of RLT buffer and β-mercaptoethanol (β-ME) at −80 °C till further processing.

Briefly, an RNA fraction highly enriched in small RNA species (≤200 nucleotides) was isolated from the leukocyte lysates using a mirVana microRNA Isolation kit (Ambion, Austin, TX, USA). The procedure is based on purification of a size fraction enriched in microRNAs. The glass-fiber filter procedure uses solutions formulated specifically for microRNA retention to avoid the loss of small RNAs that is typically seen with glass-fiber filter methods.

mRNAs of cardiovascular disease associated microRNAs (miR-1-3p, miR-16-5p, miR-17-5p, miR-20a-5p, miR-20b-5p, miR-21-5p, miR-23a-3p, miR-24-3p, miR-26a-5p, miR-29a-3p, miR-92a-3p, miR-100-5p, miR-103a-3p, miR-125b-5p, miR-126-3p, miR-130b-3p, miR-133a-3p, miR-143-3p, miR-145-5p, miR-146-5p, miR-155-5p, miR-181a-5p, miR-195-5p, miR-199a-5p, miR-210-3p, miR-221-3p, miR-342-3p, miR-499a-5p, and miR-574-3p) were reverse transcribed into complementary DNA (cDNA) using miRNA-specific stem loop primers (a part of TaqMan MicroRNA Assays) and TaqMan MicroRNA Reverse Transcription Kit (Applied Biosystems, Branchburg, NJ, USA). Reverse transcription was performed in a total reaction volume of 10 µL [52]. Subsequently, 3 µL of cDNA were mixed in a total reaction volume of 15 µL with specific primers and TaqMan MGB probes (the components of TaqMan MicroRNA Assays) and the components of the TaqMan Universal PCR Master Mix (Applied Biosystems, Branchburg, NJ, USA). Real-time RT-qPCR reactions were performed on 7500 Real-Time PCR System under standard TaqMan PCR conditions described in the TaqMan guidelines [52].

The microRNA gene expression was determined using the comparative Ct method [53]. The normalization factor [54] (geometric mean of Ct values of selected endogenous controls: RNU58A and RNU38B) was used to normalize microRNA gene expression data [52].

Selection and validation of endogenous controls for microRNA expression studies in whole peripheral blood samples affected by pregnancy-related complications has already been described in our previous study [55]. In brief, expression of 20 candidate endogenous controls (HY3, RNU6B, RNU19, RNU24, RNU38B, RNU43, RNU44, RNU48, RNU49, RNU58A, RNU58B, RNU66, RPL21, U6 snRNA, U18, U47, U54, U75, Z30 and cel-miR-39) was investigated using NormFinder [56]. RNU58A and RNU38B were identified as the most stable small nucleolar RNAs (ncRNA) and equally expressed between patients with normal and abnormal course of gestation. Therefore, these ncRNA were selected as the most suitable endogenous controls for normalization of microRNA qPCR expression studies performed on whole peripheral blood samples affected by pregnancy-related complications.

### 4.4. Statistical Analysis

Innitially, power analysis was used to determine the sample size required to detect an effect of a given size with a given degree of confidence (G*Power Version 3.1.9.6, Franz Faul, University of Kiel, Germany). 51 cases and 51 controls were needed to be tested to achieve the power 0.805. 70 cases and 70 controls were needed to be tested to achieve the power 0.902.

With respect to non-normal data distribution, the unpaired nonparametric tests were used for subsequent statistical analyses. MicroRNA gene expression was compared between preterm and term pregnancies using the Mann-Whitney test and the Kruskal-Wallis one-way analysis of variance. Afterwards, post-hoc test for the comparison among multiple groups and Benjamini-Hochberg correction for multiple comparisons were applied [57] (Table 4 and Table 5).

Box plots display the median, the 75th and 25th percentiles (the upper and lower limits of the boxes), the maximum and minimum values that are no more than 1.5 times the span of the interquartile range (the upper and lower whiskers), outliers (circles), and extremes (asterisks) (Statistica software version 9.0; StatSoft, Inc., Tulsa, OK, USA), respectively.

Receiver operating characteristic (ROC) curve analyses state the areas under the curves (AUC), the best cut-off points related sensitivities, specificities, positive and negative likelihood ratios (LR+, LR−), sensitivities at 10.0% false positive rate (FPR), respectively (MedCalc Software bvba, Ostend, Belgium). To select the optimal microRNA combinations logistic regression with subsequent ROC curve analyses were applied (MedCalc Software bvba, Ostend, Belgium). The logistic regression procedure allows to analyse the relationship between one dichotomous dependent variable and one or more independent variables. Another method to evaluate the logistic regression model makes use of ROC curve analysis. In this analysis, the power of the model’s predicted values to discriminate between positive and negative cases is quantified by the area under the ROC curve. To perform a full ROC curve analysis the predicted probabilities are first saved and next used as a new variable in ROC curve analysis. The dependent variable used in logistic regression then acts as the classification variable in the ROC curve analysis dialog box. Initially, we used enter logistic regression, when all independent variables are entered into the model in one single step, without checking. Secondly, we used backward logistic regression, when all independent variables first are entered into the model and next the non-significant variables are removed sequentially. Finally, we compared the results gained from ROC curve analyses acquired from various enter and backward logistic regression analyses and selected the best microRNA combinations.

In addition, to select optimal microRNA combinations variance inflating factor (VIF) was used to test the presence of multicollinearity in a regression model. When VIF is higher than 10, there is a significant multicollinearity and is cause for concern (MedCalc Software bvba, Ostend, Belgium).

Correlation between variables (microRNA gene expression and gestational age at birth) was calculated using the Spearman rank correlation coefficient (ρ).

### 4.5. Tissue Specificity of MicroRNAs

According to MirGeneDB 2.1 (mirgenedb.org, Tromso Research Foundation) most of cardiovascular disease associated microRNAs is usually expressed ubiquitously inclusive of placental tissue and therefore might be associated with pathogenesis of preterm delivery (Figure 7). The exception are miR-1-3p, miR-20b-5p, miR-155-5p, and miR-499a-5p, that according to MirGeneDB 2.1 (mirgenedb.org, Tromso Research Foundation) display a low expression in the placental tissue. All studied cardiovascular disease associated microRNAs were also detected in our placental sample used throughout the study. RNA fraction highly enriched for small RNAs isolated from the fetal part of one randomly selected placenta derived from gestation with normal course (the part of the placenta derived from the chorionic sac that encloses the embryo, consisting of the chorionic plate and villi) was used as a reference sample for relative quantification throughout the study. We have already verified the expression of cardiovascular disease associated microRNAs in placental tissues affected with gestational hypertension, preeclampsia, and intrauterine growth restriction [58], where altogether 150 placental samples derived from placental-related complications and 20 placental samples derived from controls were examined. All studied cardiovascular disease associated microRNAs were detected in all placental tissues involved in the study.

### 4.6. Information on MicroRNA-Gene-Biological Pathways Interactions

The MiRWalk database and the Predicted Target module were used to provide information on predicted targets of microRNAs dysregulated in the whole peripheral blood of mothers identified to be at risk of preterm delivery during the first trimester of gestation [59].

## 5. Conclusions

The minimal combination of 6 selected microRNAs (miR-16-5p, miR-21-5p, miR-24-3p, miR-133a-3p, miR-155-5p, and miR-210-3p) or maximal combination of 12 selected microRNAs (miR-16-5p, miR-20b-5p, miR-21-5p, miR-24-3p, miR-26a-5p, miR-92a-3p, miR-133a-3p, miR-145-5p, miR-146a-5p, miR-155-5p, miR-210-3p, and miR-342-3p) can predict in early stages of gestation the occurrence of preterm delivery before 37 weeks of gestation in 52.83% cases at 10.0% FPR.

Consecutive large-scale studies are needed to verify the data resulting from this pilot study. Nevertheless, hundreds of thousands of early gestational samples have to be collected to acquire adequate number of samples from pregnancies with preterm delivery. Nevertheless, it seems that microRNA biomarkers have promising diagnostical potential to be implemented into the current routine first trimester screening programme to predict preterm delivery.

## 6. Patents

National patent application—Industrial Property Office, Czech Republic (Patent n. PV 2021-562).

## Figures and Tables

**Figure 1 ijms-23-03951-f001:**
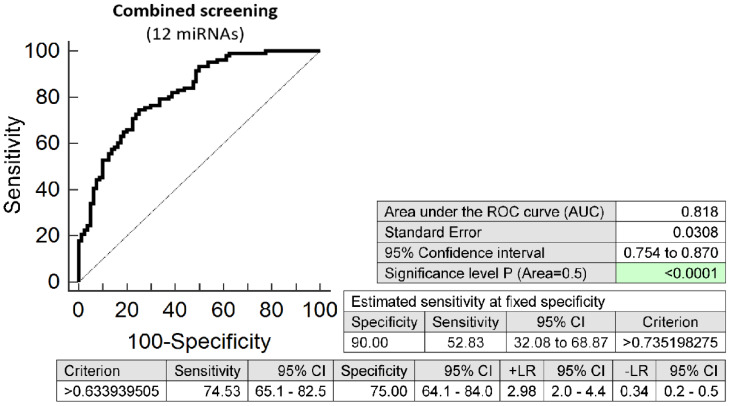
ROC analysis—the best combination of microRNA biomarkers in early prediction of preterm delivery. The combination of miR-16-5p, miR-20b-5p, miR-21-5p, miR-24-3p, miR-26a-5p, miR-92a-3p, miR-133a-3p, miR-145-5p, miR-146a-5p, miR-155-5p, miR-210-3p, and miR-342-3p showed that at 10.0% FPR 52.83% preterm pregnancies had aberrant microRNA expression profile in the whole peripheral venous blood in early stages of gestation.

**Figure 2 ijms-23-03951-f002:**
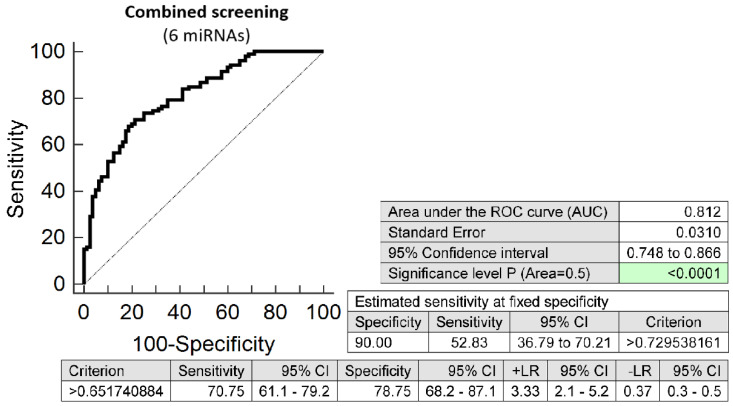
ROC analysis—the best minimal combination of microRNA biomarkers in early prediction of preterm delivery. The combination of miR-16-5p, miR-21-5p, miR-24-3p, miR-133a-3p, miR-155-5p, and miR-210-3p showed that at 10.0% FPR 52.83% preterm pregnancies had aberrant microRNA expression profile in the whole peripheral venous blood in early stages of gestation.

**Figure 3 ijms-23-03951-f003:**
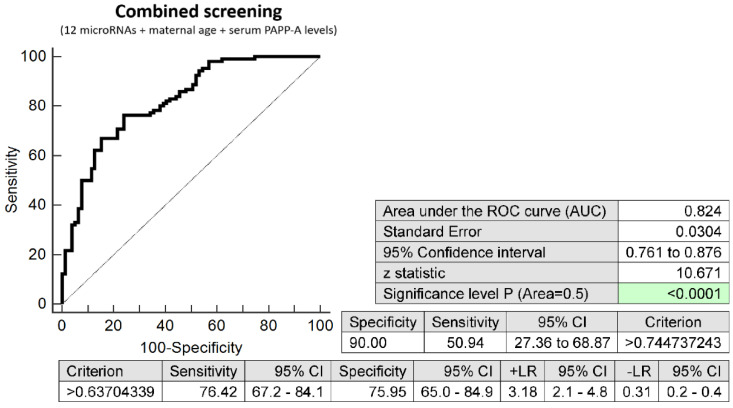
ROC analysis—combination of 12 microRNA biomarkers, maternal age at sampling and serum PAPP-A levels (IU/L) in early prediction of preterm delivery. The combination of 12 microRNAs (miR-16-5p, miR-20b-5p, miR-21-5p, miR-24-3p, miR-26a-5p, miR-92a-3p, miR-133a-3p, miR-145-5p, miR-146a-5p, miR-155-5p, miR-210-3p, and miR-342-3p), maternal age at sampling and serum PAPP-A levels (IU/L) was able to predict at 10.0% FPR the occurrence of preterm delivery in 50.94% pregnancies.

**Figure 4 ijms-23-03951-f004:**
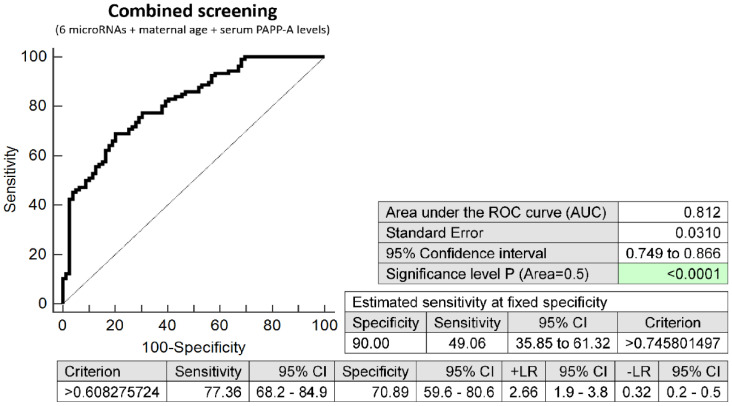
ROC analysis—combination of 6 microRNA biomarkers, maternal age at sampling and serum PAPP-A levels (IU/L) in early prediction of preterm delivery. The combination of 6 microRNAs (miR-16-5p, miR-21-5p, miR-24-3p, miR-133a-3p, miR-155-5p, and miR-210-3p), maternal age at sampling and serum PAPP-A levels (IU/L) was able to predict at 10.0% FPR the occurrence of preterm delivery in 49.06% pregnancies.

**Figure 5 ijms-23-03951-f005:**
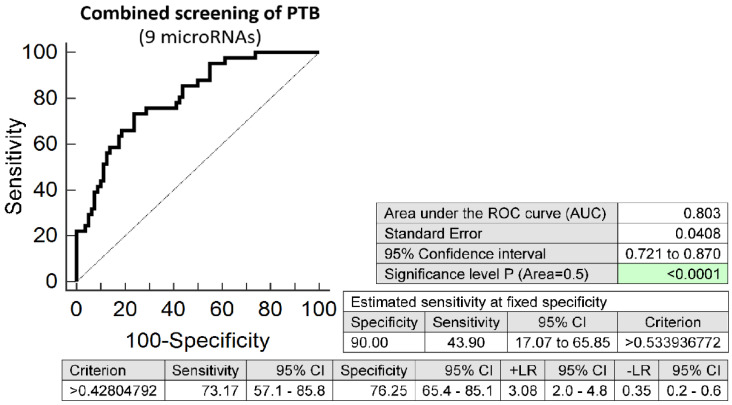
ROC analysis—the best combination of microRNA biomarkers in early prediction of spontaneous preterm birth. The combination of miR-16-5p, miR-20b-5p, miR-24-3p, miR-26a-5p, miR-92a-3p, miR-133a-3p, miR-146a-5p, miR-210-3p, and miR-342-3p showed that at 10.0% FPR 43.90% PTB pregnancies had aberrant microRNA expression profile in the whole peripheral venous blood in early stages of gestation.

**Figure 6 ijms-23-03951-f006:**
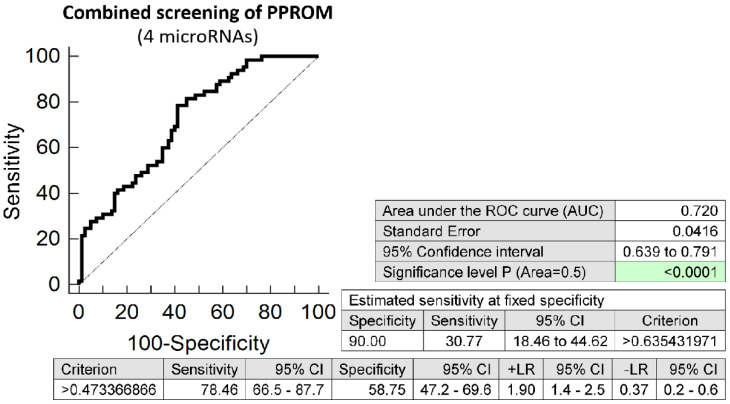
ROC analysis—the best combination of microRNA biomarkers in early prediction of preterm prelabor rupture of membranes. The combination of miR-24-3p, miR-92a-3p, miR-155-5p, and miR-210-3p showed that at 10.0% FPR 30.77% PPROM pregnancies had aberrant microRNA expression profile in the whole peripheral venous blood in early stages of gestation.

**Figure 7 ijms-23-03951-f007:**
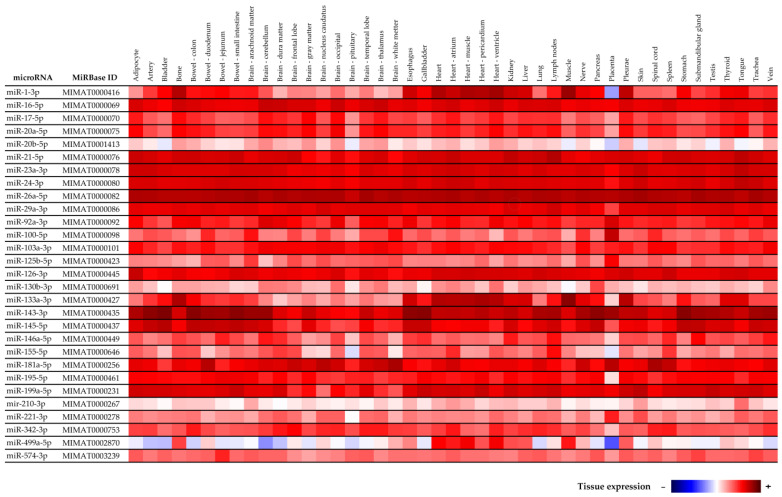
Tissue specificity of cardiovascular disease associated microRNAs according to MirGeneDB 2.1 (mirgenedb.org, accessed on 15 February 2022, Tromso Research Foundation).

**Table 1 ijms-23-03951-t001:** Clinical characteristics of the cases and controls.

	Term Normal Pregnancies (*n* = 80)	PTB (*n* = 41)	PPROM (*n* = 65)	*p*-Value ^1^	*p*-Value ^2^
** *Maternal Characteristics* **					
Maternal age (years)	32 (25–42)	33 (21–43)	32 (25–42)	0.735	1.0
Advanced maternal age (≥35 years old)	20 (25%)	15 (36.58%)	20 (30.77%)	0.183	0.439
Caucasian ethnic group	80 (100%)	41 (100%)	65 (100%)	1.0	1.0
Prepregnancy BMI (kg/m^2^)	21.28 (17.16–29.76)	22.04 (17.96–31.83)	22.14 (16.51–33.5)	1.0	1.0
Diabetes mellitus (T1DM, T2DM)	0 (0%)	3 (7.32%)	2 (3.08%)	-	-
Autoimmune diseases (SLE/APS/RA)	0 (0%)	0 (0%)	2 (3.08%)	-	-
Parity					
Nulliparous—no previous pregnancy	32 (40.0%)	11 (26.83%)	30 (46.15%)	0.152	0.456
Nulliparous—previous abortion(s) (including miscarriages)	9 (11.25%)	3 (7.32%)	12 (18.46%)	0.493	0.220
Parous—previous abortion(s) (including miscarriages)	12 (15.0%)	12 (29.27%)	11 (16.92%)	0.062	0.752
Parous—previous preterm delivery(ies)	0 (0%)	11 (26.83%)	6 (9.23%)	-	-
Parous—previous term delivery(ies)	39 (48.75%)	16 (39.02%)	17 (26.15%)	0.309	0.005
ART (IVF/ICSI/other)	2 (2.5%)	1 (2.44%)	7 (10.77%)	0.984	0.040
Smoking during pregnancy	2 (2.5%)	3 (7.32%)	2 (3.08%)	0.208	0.833
* **Pregnancy Details (First Trimester of Gestation)** *					
Gestational age at sampling (weeks)	10.29 (9.57–13.71)	10.14 (9.43–12.86)	10.14 (9.86–14.57)	0.477	0.291
MAP (mmHg)	88.75 (67.67–103.83)	88.58 (75.0–108.0)	89.58 (69.75–106.75)	1.0	1.0
MAP (MoM)	1.05 (0.84–1.25)	1.04 (0.89–1.26)	1.07 (0.83–1.26)	1.0	1.0
Mean UtA-PI	1.39 (0.56–2.43)	1.40 (0.66–2.42)	1.46 (0.69–2.95)	1.0	0.377
Mean UtA-PI (MoM)	0.90 (0.37–1.55)	0.89 (0.44–1.64)	0.91 (0.43–1.88)	1.0	0.665
PIGF serum levels (pg/mL)	27.1 (8.1–137.0)	23.65 (5.6–67.0)	24.45 (10.2–65.7)	0.327	0.245
PIGF serum levels (MoM)	1.04 (0.38–2.61)	0.96 (0.26–1.68)	0.97 (0.44–1.80)	0.347	0.468
PAPP-A serum levels (IU/L)	1.49 (0.48–15.69)	1.34 (0.34–15.28)	1.10 (0.31–8.36)	1.0	0.009
PAPP-A serum levels (MoM)	1.17 (0.37–3.18)	0.96 (0.33–3.34)	0.85 (0.21–3.24)	1.0	0.038
Free β-hCG serum levels (μg/L)	60.21 (9.9–200.6)	71.27 (17.1–234.4)	49.89 (0.96–179.78)	0.515	0.672
Free β-hCG serum levels (MoM)	1.02 (0.31–3.57)	1.19 (0.40–3.01)	0.89 (0.20–2.50)	0.850	0.248
Screen-positive for preterm delivery (<34 weeks) by FMF algorithm	5 (6.25%)	12 (29.27%)	13 (20.0%)	<0.001	0.039
Screen-positive for PE and/or FGR by FMF algorithm	0 (0%)	2 (4.88%)	8 (12.31%)	-	-
Aspirin intake during pregnancy	0 (0%)	1 (2.44%)	6 (10.77%)	-	-
** *Pregnancy Details (At Delivery)* **					
Gestational age at delivery (weeks)	40.07 (37.57–42.0)	34.57 (24.0–36.86)	35.29 (26.43–36.86)	<0.001	<0.001
Delivery at gestational age <34 weeks	0 (0%)	16 (39.02%)	13 (20.0%)	-	-
BMI (kg/m^2^)	26.66 (21.71–34.82)	25.95 (18.14–34.08)	25.86 (20.08–37.18)	0.259	0.338
Weight gain during pregnancy (kg)	14 (3–25)	10 (3–20)	11 (3–20)	<0.001	<0.001
Administration of corticosteroids	0 (0%)	23 (56.10%)	26 (40.0%)	-	-
Administration of antibiotics	0 (0%)	19 (46.34%)	52 (80.0%)	-	-
Tocolytic therapy	0 (0%)	27 (65.85%)	20 (30.77%)	-	-
Fetal birth weight (grams)	3470 (2920–4240)	2290 (680–3800)	2530 (770–3370)	< 0.001	< 0.001
Fetal sex					
Boy	40 (50.0%)	21 (51.22%)	33 (50.77%)	0.899	0.926
Girl	40 (50.0%)	20 (48.78%)	32 (49.23%)		
Mode of delivery					
Vaginal	69 (86.25%)	20 (48.78%)	46 (70.77%)	<0.001	0.022
CS	11 (13.75%)	21 (51.22%)	19 (29.23%)		
Apgar score <7, 5 min	0 (0%)	2 (4.88%)	0 (0%)	-	-
Apgar score <7, 10 min	0 (0%)	0 (0%)	0 (0%)	-	-
Umbilical blood pH	7.3 (7.29–7.38)	7.3 (6.9–7.3)	7.3 (6.9–7.31)	0.701	1.0

Continuous variables, compared using the Kruskal-Wallis test, are presented as median (range). Categorical variables, presented as number (percent), were compared using Chi-squared test. *p*-value ^1,2^: the comparison among normal pregnancies and PTB or PPROM, respectively. PTB, spontaneous preterm birth; PPROM, preterm prelabor rupture of membranes; BMI, body mass index; T1DM, type 1 diabetes mellitus; T2DM, type 2 diabetes mellitus; SLE, systemic lupus erythematosus; APS, antiphospholipid syndrome; RA, rheumatoid arthritis; ART, assisted reproductive technology; IVF, in vitro fertilization; ICSI, intracytoplasmic sperm injection; MAP, mean arterial pressure; UtA-PI, uterine artery pulsatility index; PIGF, placental growth factor; PAPP-A, pregnancy-associated plasma protein-A; β-hCG—beta-subunit of human chorionic gonadotropin; PE, preeclampsia; FGR, fetal growth restriction; FMF, Fetal Medicine Foundation; CS, Caesarean section; -, statistics cannot be performed.

**Table 2 ijms-23-03951-t002:** A List of Predicted Targets of MicroRNAs Dysregulated in the Whole Peripheral Blood of Pregnancies at Risk of Preterm Delivery in relation to the Apoptosis Pathway, using the miRWalk2.0 Database (Data Available in the KEGG, WIKI, and Panther Pathways).

	Predicted Targets		
MicroRNA	KEGG Pathways	Wiki Pathways	Panther Pathways
miR-16-5p	BCL2, IKBKB, IRAK2, PRKAR1A, PRKAR2A	BCL2, IKBKB	BCL2, CRADD, IKBKB
miR-20b-5p	ATM, CASP6, CASP7, CASP8, CASP10, CFLAR, CYCS, DFFA, EXOG, FASLG, **IL1** **R1**, IRAK1, IRAK4, MAP3K14, PIK3R2, PPP3CA, PRKAR2A, PRKX, TNFRSF10A, TNFRSF10D, XIAP	BNIP3L, CASP6, CASP7, CASP8, CASP10, CFLAR, CYCS, DFFA, FASLG, IRF1, TNFRSF1B, TNFRSF21	ATF6, BAG1, CASP7, CASP8, CASP10, CFLAR, CREB1, CREM, CYCS, EIF2S1, FASLG, HSPA5, MAP3K14, MAPK9, PRKCQ, REL, TNFRSF10D, TNFRSF1B, XIAP
miR-21-5p	APAF1, CFLAR, FASLG	APAF1, CFLAR, FASLG, MAP3K1	APAF1, CFLAR, DAXX, EIF2S1, FASLG, MAP2K3
miR-24-3p	BCL2L1, EXOG, FASLG, IKBKB, IL1B, IRAK4, MYD88, PIK3CB, RIPK1	BBC3, BCL2L2, BCL2L11, BNIP3L, FASLG, IKBKB, MYC, NFKBIE, RIPK1, TRAF1, TRAF3	BCL2L1, BCL2L2, BCL2L11, EIF2AK2, FASLG, FOS, IKBKB, PIK3CB, PRKCA, PRKCH, RIPK1
miR-26a-5p	APAF1, BID, BIRC2, CASP6, DFFB, PPP3CB, PPP3CC	APAF1, BAK1, BID, BIRC2, CASP6, CRADD, DFFB, MDM2, **PMAIP1**	APAF1, ATF2, BAG4, BAK1, BID, BIRC2, CRADD, CREB1, EIF2AK2, FOS, HSPA8, PRKCD, PRKCQ, RELB
miR-92a-3p	APAF1, ATM, BCL2L1, BCL2L11, BIRC3, CASP10, CASP8, CASP9, CFLAR, CSF2RB, CTSB, CTSS, CTSV, DAB2IP, DFFA, DFFB, FOS, GZMB, HRK, KRAS, MAP3K14, MAPK10, NTRK1, RIPK1, TNFRSF10B, TP53, XIAP	AKT1, APAF1, BCL2L1, BCL2L11, BCL2L2, BIRC3, CASP10, CASP8, CASP9, CFLAR, DFFA, DFFB, GZMB, IGF1, IGF2, IRF1, IRF4, MAPK10, MDM2, RIPK1, TNFRSF10B, TNFRSF25, TP53, TP63, TP73, TRAF3	APAF1, APPL1, BCL2L1, BCL2L11, BMF, CARD8, CASP8, CASP9, CDH1, CFLAR, DAPK2, DAPK3, DFFA, DFFB, GZMB, KPNB1, MAPT, NMT1, PAK2, PSMB2, PSMD5, PSMD8, PSMD9, PSMF1, PTK2, RIPK1, STAT3, STK26, **TFDP2**, TNFRSF10B, TP53, TP63, TP73, UNC5B, XIAP, YWHAZ
miR-126-3p	TNFRSF10B	TNFRSF10B	-
miR-133a-3p	ENDOD1, **IRAK3**, MAP3K14, TNFRSF10B	BCL2L2, BNIP3L, TNFRSF10B	BCL2L2, MAP3K14, TNFRSF10B
miR-145-5p	AIFM1, PIK3R5, TNFRSF10B	TNFRSF10B, TNFRSF25	AIFM1, MAP4K2, TMBIM6, TNFRSF10B
miR-146a-5p	CASP7, CASP9, DFFA, IL3, IRAK1, IRAK4, PPP3R2, PRKACA	CASP2, CASP7, CASP9, DFFA, **PMAIP1**, PRF1	BAG1, CASP7, CASP9, HSPA1A, JDP2, PRKCE
miR-155-5p	PIK3R1, LMNB2	AKT1, IGF1, PIK3R1	PSMD8, TJP1
miR-210-3p	AKT1, ATM, BAK1, BBC3, BCL2L11, BIRC3, CASP10, CSF2RB, CTSB, EIF2S1, ERN1, MAPK10, MCL1, PDPK1, RAF1, RIPK1, TP53AIP1	AKT1, BAK1, BBC3, BCL2L11, BIRC3, CASP10, IGF1, IGF1R, IRF2, MAPK10, MCL1, RIPK1, TP63, TP73	AKT1, APPL1, BAK1, BBC3, BCAP31, BCL2L11, CLSPN, DAPK1, GSDME, HMGB1, MAPT, PAK2, PSMA5, PSMB2, PSMB5, PSMC1, PSMD9, PSME3, PTK2, RIPK1, ROCK1, STAT3, **TFDP2**, TP63, UBA52
miR-221-3p	AKT3, APAF1, CASP10, IKBKG, **IL1RAP**, PIK3CD, PPP3R1, TNFSF10	APAF1, BNIP3L, CASP10, IKBKG, IRF4, MAPK10, MDM2, TNFSF10	AKT3, APAF1, ATF2, ATF4, CASP10, CREB1, MAPK10, PIK3CD, PRKCB, TNFSF10
miR-342-3p	ACTG1, CFLAR, EIF2S1, MAPK1, MAPK9, **PMAIP1**	BCL2L2, CFLAR, **PMAIP1**	CFLAR, DAPK2, DYNLL2, E2F1, MAPK1, OCLN, **PMAIP1**, PSME3, ROCK1, SFN, TFDP2, TICAM2, UNC5B

Meta-analysis of maternal and fetal transcriptomic data revealed in total 210 differentially expressed genes (65 genes upregulated and 145 genes downregulated) in maternal peripheral blood during the third trimester of gestation in PTB pregnancies, in which case half of these genes were immune related. Dysregulated genes were highly involved in the innate and adaptive immune responses. For example, within upregulated genes IL1R1, IL1RAP, and IRAK 3 were identified. In addition, within downregulated genes TFDP2 and PMAIP1 were identified [36]. The dysregulated genes identified in meta-analysis of maternal and fetal transcriptomic data are marked in bold; - no predicted targets.

**Table 3 ijms-23-03951-t003:** A List of Predicted Targets of MicroRNAs Dysregulated in the Whole Peripheral Blood of Pregnancies at Risk of Preterm Delivery in relation to the Senescence and Autophagy Pathways and the Inflammatory Response Pathway, using the miRWalk2.0 Database (Data Available in WIKI Pathways).

	Predicted Targets (Wiki Pathways)	
MicroRNA	Senescence and Autophagy Pathways	Inflammatory Response Pathway
miR-16-5p	BCL2, CREG1, HMGA1, LAMP2, MAP2K1, RAF1, SMAD4	IL2RA
miR-20b-5p	ATG10, ATG12, CD44, CDKN1A, E2F1, IL6R, IRF1, LAMP2, RNASEL, RSL1D1, SERPINE1, SH3GLB1	**CD28**, IL5, LAMC1, LAMC2, TNFRSF1B
miR-21-5p	MAP2K3	THBS3
miR-24-3p	ATG13, CDKN1B, FN1, IFNG, IGFBP5, IL1B, IL6R, MAP1LC3A, MAP1LC3C, MMP14	**CD28**, CD86, FN1, IFNG, IL2RB, LAMC1
miR-26a-5p	ATG13, COL10A1, HMGA1, IFNG, IL6, MDM2, PCNA, PTEN, RB1, ULK1	COL1A2, IFNG
miR-92a-3p	CXCR2, HMGA1, IGF1, IL24, IL6R, IL6ST, ING1, IRF1, LAMP2, MAP2K6, MDM2, MMP14,PIK3C3, PLAT, SLC39A1, TP53	**CD28**, CD40, IL2RA, IL5RA
miR-126-3p	-	-
miR-133a-3p	ATG14, FN1, GABARAPL1, MAPK14, MMP14, RB1CC1, SLC39A1	**CD28**, FN1
miR-145-5p	AMBRA1, CD44, HMGA1, IFNB1, MAP1LC3B, SLC39A2	IL2RA
miR-146a-5p	ATG12, IL3, KMT2A, RNASEL, SERPINB2, TNFSF15	CD80, CD86
miR-155-5p	ATG7, BRAF, CDK6, IGF1	**CD28**
miR-210-3p	CDC25B, IGF1R, INHBA, MAPK14, PTEN, RAF1, SMAD3, SMAD4, IGF1, MAP2K6, PIK3C3	**CD28**, IL5RA
miR-221-3p	-	THBS1, VTN
miR-342-3p	ATG14, ATG16L1, E2F1, GSK3B, MAPK1, ULK1, ING1	-

Meta-analysis of maternal and fetal transcriptomic data revealed in total 210 differentially expressed genes (65 genes upregulated and 145 genes downregulated) in maternal peripheral blood during the third trimester of gestation in PTB pregnancies, in which case half of these genes were immune related. Dysregulated genes were highly involved in the innate and adaptive immune responses. For example, within downregulated genes CD28 was identified [36]. The dysregulated genes identified in meta-analysis of maternal and fetal transcriptomic data are marked in bold. - no predicted target.

**Table 4 ijms-23-03951-t004:** Benjamini-Hochberg correction: Comparison of microRNA gene expression between preterm (delivery before 37 weeks of gestation) and normal term pregnancies (delivery after 37 weeks of gestation).

K	i	Alpha = 0.05	Alpha = 0.01	Alpha = 0.001
**2**		**0.05**	**0.01**	**0.001**
	**1**	0.025	0.005	0.001

**Table 5 ijms-23-03951-t005:** Benjamini-Hochberg correction for multiple comparisons: Comparison of microRNA gene expression between preterm and normal term pregnancies (PTB vs. PPROM vs. normal pregnancies).

K	i	Alpha = 0.05	Alpha = 0.01	Alpha = 0.001
**3**		**0.05**	**0.01**	**0.001**
	**1**	0.017	0.003	0.000
	**2**	0.033	0.007	0.001
	**3**	0.050	0.010	0.001

## Data Availability

The data presented in this study are available on request from the corresponding author. The data are not publicly available due to rights reserved by funding supporters.

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
