# Peer review of "First Trimester Prediction of Preterm Delivery in the Absence of Other Pregnancy-Related Complications Using Cardiovascular-Disease Associated MicroRNA Biomarkers"

_ijms, 2022, doi:10.3390/ijms23073951_

Round 1
Reviewer 1 Report
This manuscript by Hromadnikova et al. and colleagues demonstrated the potential of cardiovascular diseases associated miRNA expression from maternal leukocytes during early pregnancy in predicting preterm delivery. Although it’s an interesting study there are several points that need to be addressed to improve the quality of the manuscript.
Major Comments:
- The main limitation of this manuscript is that I don’t think it falls within the aims/scope of the IJMS (advanced forum for molecular studies in biology and chemistry, with a strong emphasis on molecular biology and molecular medicine…). There was no study linking the reason as to why miRNAs were decreased or the direct biological effect of decrease miRNA detected within leukocytes that could contribute to preterm birth.
- Could the postulate the expression of these miRNAs in placenta samples of Term Normal, PTB, and PPROM samples? are these miRNAs have any tissue specificity to be claimed as biomarkers for preterm delivery.
- Several reports suggest that some of these proposed miRNAs (eg., miR-155-5p, miR-21-5p, and miR-146-5p) were inflammatory in nature and their upregulation led to apoptosis. Could the authors justify why these miRNAs were downregulated in maternal leukocytes during preterm deliveries?
- Study design and methods:
- The author described a case-control retrospective study using samples matched to gestation age at collection and sample storage time. From the number of participants in Table 1, it seems as though it was not a 1:1 match. Can the author clarify this, and if so, did the author put this into consideration during statistical analysis?
- Sample size needs to be clarified – in the text author stated 29 pregnancies delivered before 34 weeks and 77 after 34 weeks. Was the “77 after 34 weeks” referring to PTL between 34-37 weeks?
- Sample processing: the article studied “miRNA in peripheral blood leukocyte”, but the sample processing stated “small RNAs was isolated from the whole peripheral venous blood”. Was there a mistake or author is presuming the small RNA isolated from whole blood represents what is seen in leukocytes? If so I don’t think that is accurate.
- Was correlation between the 12 miRNA expressions examined before combining them to assess and avoid multicollinearity? Collinearity of miRNA may explain the same predictive capability of 12 miRNA vs. 6 miRNA, especially given that the ROC curve % if the 6 excluded miRNA are between 20-22%.
- The datasets present an opportunity also to examine the correlation between downregulation of miR vs. GA at birth. Was the relationship between miRNA and GA at birth done? Given that the author has over 100 patients with a GA spread at delivery, this may be informative.
- Instead of using combinations of miRNA alone, authors should also examine if including other clinical variables (i.e., Age, maternal hx) and biomarkers (e.g., PAPP-A) along with the miRNA combinations would result in a better prediction model.
Minor comment:
- Abstract need some reformatting – example:
- The beginning sounds very odd like they are trying to cut words and save space
- Line 26 – “was associated with preterm delivery after 34 weeks of gestation” better to be “was associated with preterm delivery between 34+0/7 weeks to 36+6/7 week
- Line 28 – “had also” à “also had”
- Clarification – PPROM usually refers to preterm premature rupture of membrane, ie, ROM before 37 weeks.
- I don’t think the asterisk sign following the P-value is necessary and makes paper hard to read.
Author Response
"Please see the attachment."

Reviewer 2 Report
- Materials and Methods, Line 299-301: You wrote, “Forty-one out of 4469 pregnancies were confirmed to have diagnosis of spontaneous preterm birth, and 65 pregnancies were diagnosed with preterm prelabor rupture of membranes.” If so, the incidence rate of spontaneous preterm birth was 0.92%, and the incidence rate of pPROM was 1.45%. I think that the incidence rates of PB or pPROM were too small. How do you explain the facts? I would know the incidence rate of all preterm birth in your country.
- Materials and Methods, Line 310: You should move the description and location of Table 1 to the Results, the first paragraph.
- Materials and Methods, Line 311-316: This is a retrospective study, but not prospective study. How and when did you gain informed consent from all the participants? Usually, opt out is performed in the retrospective study.
- Results: You did not show how you selected the combination of miRs. You should write how to select the pairs in Logistic regression analysis in Materials and Methods, Statistics section.
- Discussion: In this study, you used peripheral blood leukocytes. However, plasma is usually used for the prediction of pregnancy outcomes. Why did not use the plasma for the detection of miRs? You should discuss the usage of peripheral blood leukocytes but not plasma.
Author Response
"Please see the attachment."

Round 2
Reviewer 1 Report
The authors have carefully edited the manuscript to clarify the content and improve the overall scientific work.
Author Response
Thank you for positive evaluation of our research. We appreciate it.
